# Disparities and trends of the incidence and mortality of female-specific cancers in the United States

**Ze Zhang**[1☉], **Yunhai Li**[1☉], **Hongbo Huang**[1], **Tingting Wei**[1], **Ying Huang**[1], **Xiuquan Qu**[1], **Yijing Xu**[1], **Aijie Zhang**[2], **Jiaying Li**[1], **Zheng Gong**[1], **Zhiqi Hu**[1], **Fan Li**[1]*

**1** Department of Breast and Thyroid Surgery, Chongqing Key Laboratory of Molecular Oncology and Epigenetics, The First Affiliated Hospital of Chongqing Medical University, Chongqing, China, **2** University-Town Hospital of Chongqing Medical University, Chongqing, China

☉ These authors contributed equally to this work.
* 202899@hospital.cqmu.edu.cn

## Abstract

### Background

Female-specific cancers (FSCs) impose substantial burdens on healthcare systems and economies worldwide. The significant impact of these cancers in the United States warrants further investigation.

### Objective

This study aimed to analyze trends in incidence and mortality rates of six female-specific cancers (breast, cervical, uterine, ovarian, vaginal, and vulvar cancers) among diverse racial and ethnic groups in the United States, and to evaluate the attributable contributions of major risk factors to the cancer death burden as well as their temporal changes.

### Methods

Incidence and mortality data were obtained from the SEER (22 registry) database, to examine cross-sectional and temporal trends by race/ethnicity. The burden of FSCs attributable to specific risk factors was estimated based on the Global Burden of Disease 2021 database.

### Results

Between 2017 and 2021, Breast cancer incidence increased across all racial groups, most notably among non-Hispanic White (White) women, while mortality declined. Cervical cancer incidence decreased in most groups but remained stable among American Indians and Alaska Native (AIAN) women. Uterine cancer incidence increased across all racial groups, except for Whites. Incidence and mortality rates

**Data availability statement:** The data utilized in this study can be found in the Surveillance, Epidemiology, and End Results database Public-access SEER data (https://seer.cancer.gov/), National Center for Health Statistics (NCHS), and 2021 Global Burden of Disease (GBD) study (https://vizhub.healthdata.org/gbd-results/). Analytic code and any additional information required to reanalyze the data reported in this paper can be obtained from the corresponding author upon reasonable request, following the publication of this article.

**Funding:** This study was supported by the National Natural Science Foundation of China (grant 82202913 and 82372996) and the Natural Science Foundation of Chongqing (grant CSTB 2023 NSCQ-MSX0480). The funders had no role in study design, data collection and analysis, decision to publish, or preparation of the manuscript.

**Competing interests:** The authors have declared that no competing interests exist.

for ovarian and vaginal cancers remained stable or decreased, whereas vulvar cancer mortality was highest among White and AIAN women and lowest among non-Hispanic Asian Americans and Pacific Islander (AAPI) women. From 2000 to 2021, risk-attributable deaths decreased for breast, cervical, and ovarian cancers but increased for uterine cancer.

## Conclusions

Significant sociodemographic disparities and unfavorable trends persist in the incidence and mortality of all six major female-specific cancers, highlighting the importance for effective prevention and intervention strategies.

## 1. Introduction

According to pertinent research reports, the four most common cancers among women worldwide are breast, colorectal, lung cancer, and cervical cancer, accounting for 48.8% of all cancers [1]. Female-specific cancers (FSCs)—malignancies of the breast and female reproductive system, including breast, cervical, uterine, ovarian, vaginal, and vulvar cancers—representing 44% of all cancer cases in women [1,2]. In 2024, an estimated 427,650 new cases of female-specific cancers and 76,100 related deaths are projected in the United States, accounting for approximately 44% of all new cancer cases and 26% of cancer deaths among women [2]. With a growing and aging population, these numbers are expected to rise, further straining the healthcare system and increasing the economic burden. For example, in 2019, breast cancer alone incurred costs of $4.25 billion. [3].

Significant racial and ethnic disparities persist in the burden of FSCs in the US, strongly linked to social determinants of health (SDOH)—including economic stability, education access, health care quality, neighborhood environment, and social context—which disproportionately affect disadvantaged populations, including racial/ethnic minorities, low-income groups, and individuals with limited healthcare access [4]. For instance, while non-Hispanic White women historically have higher breast cancer incidence, Black women experience disproportionately higher mortality [5]. Most existing research has often focused on individual cancer types rather than the comprehensive burden of all FSCs. Moreover, there is a critical gap in recent, systematic analyses of temporal trends in FSC incidence and mortality, particularly across racial/ethnic groups and in relation to SDOH.

To address these gaps, this study comprehensively assesses the burden of FSCs in the US by examining sociodemographic disparities and trends in incidence and mortality, incorporating an analysis of associated risk factors. The findings provide valuable insights for healthcare policymakers, funding agencies, and researchers to develop effective strategies aimed at reducing the impact of FSCs, particularly among vulnerable groups.

## 2. Materials and methods

### 2.1. Data source

To analyze cross-sectional incidence rates from 2017 to 2021 and short-term (2017−2021) and long-term (2000−2021) trends, we obtained incidence data from the National Cancer Institute (NCI) Surveillance, Epidemiology, and End Results (SEER) 22-registry database, which encompasses 47.9% of the U.S. population [6]. Cancer sites were classified using the International Classification of Diseases for Oncology, Third Edition (ICD-O-3) [7]. The SEER database uses the following ICD-O-3 codes: breast cancer (C50.0–C50.9), cervical cancer (C53.0–C53.9), uterine cancer (C54.0–C55.9), ovarian cancer (C56.9), vaginal cancer (C52.9), and vulvar cancer (C51.0–C51.9).

Mortality data from 2000 to 2022 were obtained from the SEER mortality database, compiled by the National Center for Health Statistics (NCHS), which covers nearly 100% of the US population [8]. Causes of death were coded according to ICD-10 [7]. Only primary cancers were considered: breast (C50), cervical (C53), uterine (C54–C55), ovarian (C56), vaginal (C52), and vulvar (C51) cancers [8].

To further assess the cancer burden attributable to risk factors, we sourced specific data from the Global Burden of Disease (GBD) 2021 database. The GBD estimates are informed by vital registration systems, verbal autopsy reports, and population-based cancer registries [9]. Four major FSCs were classified according based on ICD-10 codes [10].

The incidence and mortality rates with their 95% confidence intervals were calculated by distinct racial and ethnic groups as documented in medical records or death certificates non-Hispanic White (White), non-Hispanic Black (Black), non-Hispanic Asian American and Pacific Islander (AAPI), non-Hispanic American Indian and Alaska Native (AIAN), Hispanic, and all races/ethnicities and place of residence (state) [11].

### 2.2. Statistical analysis

Age-adjusted incidence and mortality rates were calculated using SEER*Stat version 8.4.4 [12] and expressed per 100,000 persons. Incidence rates were adjusted for delays in reporting. Rate comparisons between groups were considered statistically significant if the 95% confidence intervals, calculated using the Tiwari method [13], did not include zero. The Tiwari method reduces bias when comparing rates across regions with varying population sizes [13]. Rate ratios for different racial/ethnic groups were also calculated using this method [11].

To distinguish between enduring patterns and recent fluctuations, we analyzed both long-term and short-term trends. Temporal trends in age-standardized cancer incidence (2000–2021) and mortality (2000–2022) rates were estimated using the Joinpoint Trend Analysis Software (Version 4.9.0). Joinpoint regression identifies significant inflection points where trends change in direction or magnitude, calculating the annual percent change (APC) for each segment [14]. The default number of Joinpoints ranged between 0 and 5, with the estimate best-fitting model selected via Monte Carlo permutation analysis [14]. The Monte Carlo permutation analysis is a statistical method to detect significant changes in trends by randomly permuting the data and comparing different models [14]. It identifies the best-fitting model by testing whether observed Joinpoints are significantly different from what would occur by chance [14]. This approach controls for Type I error and is flexible, making it reliable for analyzing complex cancer incidence and mortality trends [14]. Short-term trends in incidence (2017–2021) and mortality (2018–2022) were expressed as the average annual percent change (AAPC) [11]. The AAPC was equal to the APC when the AAPC was entirely within the last Joinpoint segment. Trends were characterized as increasing or decreasing when the APC or AAPC was statistically significant (two-sided p < 0.05); otherwise, they were considered stable [7]. When the numbers of cases or deaths used to compute rates are small, those rates tend to have poor reliability. Therefore, to discourage misinterpretation or misuse of rates or counts that are unstable, incidence and death rates and counts are not shown in tables and figures when the case or death counts are below 16 [15].

To illustrate the geographic disparities in contemporary incidence and mortality rates, we utilized R software (version 4.4.2) with data from the SEER cancer statistics to generate maps of age-standardized incidence rates (2017–2021) and

mortality rates (2018–2022) by state. Due to data limitations, mapping was restricted to breast, cervical, uterine, and ovarian cancers.

We utilized the GBD 2021 risk factor data for four FSCs (breast, cervical, uterine, and ovarian cancers), to analyze temporal trends in risk-attributable burden using standard epidemiological methods.

### 2.3. Ethical statement

As this study used publicly available summary data from the Surveillance, Epidemiology, and End Results database Public-access SEER data, National Center for Health Statistics (NCHS), and 2021 Global Burden of Disease (GBD) study, ethical approval was not required.

## 3. Results

### 3.1. Incidence and mortality by race and ethnicity

Table 1 presents the cross-sectional incidence (2017–2021) and mortality (2018–2022) rates for six major female-specific cancers (FSCs) by race/ethnicity.

Among all groups, breast cancer had the highest incidence rate (129.4; 95% CI, 129.1–129.8 per 100,000), while vaginal cancer had the lowest incidence rate (0.7; 95% CI, 0.6–0.7) between 2017 and 2021. The incidence rate of breast cancer was highest among non-Hispanic White (White) (139.0; 95% CI, 138.5–139.4), followed by non-Hispanic Black (Black), non-Hispanic American Indians and Alaska Native (AIAN), non-Hispanic Asian Americans and Pacific Islander (AAPI), and Hispanic. The incidence rate of cervical cancer was highest among Hispanics (9.8; 95% CI, 9.6–10.0) and lowest among AAPIs (6.1; 95% CI, 5.9–6.4). Blacks had the highest incidence rate of uterine cancer (30.2; 95% CI, 29.7–30.6) and vaginal cancer (0.9; 95% CI, 0.8–1.0), while AAPIs had the lowest risk of uterine and vaginal cancers, with an incidence ratio of (23.4; 95% CI, 23.0–23.9) and (0.4; 95% CI, 0.3–0.4), respectively. Ovarian cancer exhibited the highest incidence rate among AIANs (11.6; 95% CI, 9.9–13.5) with comparable developing risk with Whites Rate ratio (RR), (1.12; 95% CI, 0.95–1.30). The highest incidence rate of vulvar cancer was observed among Whites (3.1; 95% CI, 3.0–3.1), which was similar with AIANs (3.0; 95% CI, 2.1–4.0), whereas the Black, AAPI, and Hispanic exhibited lower incidence rates and risks compared to Whites.

Blacks have the highest mortality rates for breast (26.8; 95% CI, 26.5–27.1), cervical (3.2; 95% CI, 3.1–3.3) and uterine (9.5; 95% CI, 9.4–9.7) cancers among all ethnic groups between 2018 and 2022, while the lowest rates were observed in AAPIs, with a mortality rate of 11.9 (95% CI, 11.6–12.1), 1.6 (95% CI, 1.5–1.7), and 3.7 (95% CI, 3.5–3.8), respectively. The mortality rate for ovarian cancer was highest among Whites (6.3; 95% CI, 6.3–6.4), followed by AIAN, Black, Hispanic, and AAPI. The mortality rates of vaginal cancer were comparable between White, Black, Hispanic. In terms of vulvar cancer, the mortality rates were higher among individuals of Whites (0.7; 95% CI, 0.7–0.7) and AIANs (0.6; 95% CI, 0.4–0.9). Conversely, the lowest mortality rate of vulvar cancer was observed among individuals of AAPI (0.2; 95% CI, 0.2–0.2).

### 3.2. Temporal trends in incidence and mortality

From 2000 to 2021, breast cancer incidence rates increased across all races (average annual percentage change (AAPC), 1.4; 95% CI, 0.7 to 2.5), with particularly notable increased among AAPIs and AIANs (Fig 1A, Table 2).

Concurrently, mortality rates of breast cancer exhibited a declining trend (AAPC, −1.5; 95% CI, −2.0 to −1.3) across all races, with the most significant reduction observed among Blacks (AAPC, −1.4; 95% CI, −1.5 to −1.3), although their mortality rate remains the highest, while mortality rates remained stable among AAPIs (AAPC, 0.3; 95% CI, −0.1 to 2.1) and Hispanic (AAPC, −0.7; 95% CI, −0.9 to 0.8) (Fig 2, Table 3).

**Table 1. Average annual age-standardized incidence (2017–2021) and death (2018–2022) rates and rate ratios with 95% confidence intervals for major female-specific cancers by race and ethnicity in the USA.**

| | White | Black | AAPI | AIAN | Hispanic | All races |
|---|---|---|---|---|---|---|
| **Incidence** | | | | | | |
| **Breast cancer** | | | | | | |
| Rate (1/100000) | 139.0 (138.5-139.4) | 129.3 (128.3-130.3) | 110.3 (109.3-111.3) | 113.0 (107.7-118.6) | 101.2 (100.5-101.9) | 129.4 (129.1-129.8) |
| Rate ratio | Reference | 0.93 (0.92-0.94)* | 0.79 (0.79-0.80)* | 0.81 (0.77-0.85)* | 0.73 (0.72-0.73)* | — |
| Count | 397265 | 68202 | 47762 | 1795 | 84177 | 604721 |
| **Cervical cancer** | | | | | | — |
| Rate (1/100000) | 6.9 (6.8-7.0) | 8.7 (8.4-8.9) | 6.1 (5.9-6.4) | 9.2 (7.7-10.9) | 9.8 (9.6-10.0) | 7.6 (7.6-7.7) |
| Rate ratio | Reference | 1.26 (1.22-1.31)* | 0.09 (0.85-0.93)* | 1.33 (1.12-1.58)* | 1.43 (1.39-1.47)* | — |
| Count | 15176 | 4366 | 2592 | 134 | 8614 | 31234 |
| **Uterine cancer** | | | | | | |
| Rate (1/100000) | 27.7 (27.5-27.9) | 30.2 (29.7-30.6) | 23.4 (23.0-23.9) | 29.1 (26.5-32.0) | 26.9 (26.6-27.3) | 28.0 (27.9-28.2) |
| Rate ratio | Reference | 1.08 (1.07-1.11)* | 0.84 (0.83-0.86)* | 1.05 (0.96-1.16) | 0.97 (0.96-0.99)* | — |
| Count | 85350 | 16988 | 10418 | 467 | 22941 | 137551 |
| **Ovarian cancer** | | | | | | |
| Rate (1/100000) | 10.4 (10.3-10.6) | 8.9 (8.7-9.2) | 9.4 (9.1-9.7) | 11.6 (9.9-13.5) | 10.0 (9.8-10.2) | 10.2 (10.2-10.3) |
| Rate ratio | Reference | 0.85 (0.83-0.88)* | 0.90 (0.87-0.93)* | 1.12 (0.95-1.30) | 0.96 (0.93-0.98)* | — |
| Count | 30222 | 4716 | 4051 | 181 | 8414 | 47945 |
| **Vaginal cancer** | | | | | | |
| Rate (1/100000) | 0.6 (0.6-0.7) | 0.9 (0.8-1.0) | 0.4 (0.3-0.4) | — | 0.7 (0.6-0.7) | 0.7 (0.6-0.7) |
| Rate ratio | Reference | 1.42 (1.28-1.57)* | 0.59 (0.50-0.70)* | — | 1.02 (0.92-1.12) | — |
| Count | 2007 | 482 | 163 | — | 508 | 3194 |
| **Vulvar** | | | | | | |
| Rate (1/100000) | 3.1 (3.0-3.1) | 1.9 (1.8-2.0) | 1.0 (0.9-1.1) | 3.0 (2.1-4.0) | 1.9 (1.8-2.0) | 2.6 (2.5-2.6) |
| Rate ratio | Reference | 0.61 (0.57-0.66)* | 0.32 (0.29-0.35)* | 0.97 (0.68-1.29) | 0.60 (0.57-0.64)* | — |
| Count | 9432 | 975 | 425 | 46 | 1394 | 12396 |
| **Mortality** | | | | | | |
| **Breast cancer** | | | | | | |
| Rate (1/100000) | 19.3 (19.2-19.5) | 26.8 (26.5-27.7) | 11.9 (11.6-12.1) | 17.8 (16.5-19.1) | 13.7 (13.5-13.9) | 19.3 (19.2-19.4) |
| Rate ratio | Reference | 1.38 (1.36-1.40)* | 0.61 (0.60-0.63)* | 0.92 (0.85-0.99)* | 0.70 (0.69-0.72)* | — |
| Count | 154,313 | 32,089 | 7,297 | 778 | 16,432 | 211,539 |
| **Cervical cancer** | | | | | | |
| Rate (1/100000) | 2.1 (2.0-2.1) | 3.2 (3.1-3.3) | 1.6 (1.5-1.7) | 3.0 (2.5-3.6) | 2.4 (2.3-2.5) | 2.19 (2.6-2.22) |
| Rate ratio | Reference | 1.53 (1.48-1.59)* | 0.79 (0.74-0.84)* | 1.43 (1.19-1.71)* | 1.15 (1.10-1.19)* | — |
| Count | 13,033 | 3,740 | 985 | 132 | 3,006 | 20,979 |
| **Uterine cancer** | | | | | | |
| Rate (1/100000) | 4.7 (4.7-4.8) | 9.5 (9.4-9.7) | 3.7 (3.5-3.8) | 4.7 (4.1-5.4) | 4.4 (4.3-4.5) | 5.2 (5.2-5.3) |
| Rate ratio | Reference | 2.00 (1.97-2.05)* | 0.77 (0.74-0.80)* | 1.00 (0.87-1.15) | 0.93 (0.90-0.96)* | — |
| Count | 39,734 | 12,122 | 2,278 | 216 | 5,232 | 59,760 |
| **Ovarian cancer** | | | | | | |
| Rate (1/100000) | 6.3 (6.3-6.4) | 5.5 (5.4-5.6) | 4.3 (4.2-4.5) | 6.2 (5.4-6.9) | 4.8 (4.7-4.9) | 6.0 (6.0-6.1) |
| Rate ratio | Reference | 0.87 (0.84-0.89)* | 0.68 (0.66-0.71)* | 0.98 (0.86-1.10) | 0.76 (0.74-0.78)* | — |
| Count | 51,588 | 6,808 | 2,677 | 276 | 5,750 | 67,275 |
| **Vaginal cancer** | | | | | | |
| Rate (1/100000) | 0.19 (0.18-0.20) | 0.2 (0.2-0.3) | 0.11 (0.09-0.15) | — | 0.2 (0.1-0.2) | 0.18 (0.18-0.19) |
| Rate ratio | Reference | 1.25 (1.09-1.42)* | 0.61 (0.47-0.78) | — | 0.82 (0.69-0.96)* | — |

*(Continued)*

**Table 1.** (Continued)

|  | White | Black | AAPI | AIAN | Hispanic | All races |
|---|---|---|---|---|---|---|
| Count | 1,581 | 285 | 69 | — | 169 | 2121 |
| **Vulvar cancer** |  |  |  |  |  |  |
| Rate (1/100000) | 0.72 (0.70-0.74) | 0.4 (0.36-0.44) | 0.2 (0.16-0.24) | 0.6 (0.4-0.9) | 0.4 (0.3-0.4) | 0.62 (0.61-0.64) |
| Rate ratio | Reference | 0.56 (0.51-0.61)* | 0.28 (0.23-0.33)* | 0.86 (0.57-1.29) | 0.52 (0.47-0.57)* | — |
| Count | 6,071 | 472 | 116 | 26 | 411 | 7,115 |

aAll racial groups are exclusive of individuals identifying as Hispanic. Rates are per 100000 and age-adjusted to the 2000 US standard population. Estimates based on less than 16 cases are suppressed and not shown. Incidence data are from the National Cancer Institute Surveillance, Epidemiology and End Results database, and mortality data are from the National Center for Health Statistics database.

bRates are adjusted for delays in case reporting.

cRate ratios are based on rates age-adjusted to the 2000 US standard population, with White population as the reference group.

*tSignificant difference in rate ratio in comparison to the reference group (*p* < 0.05).

Abbreviation: AIAN = American Indian and Alaska Native; AAPI = Asian American and Pacific Islander.

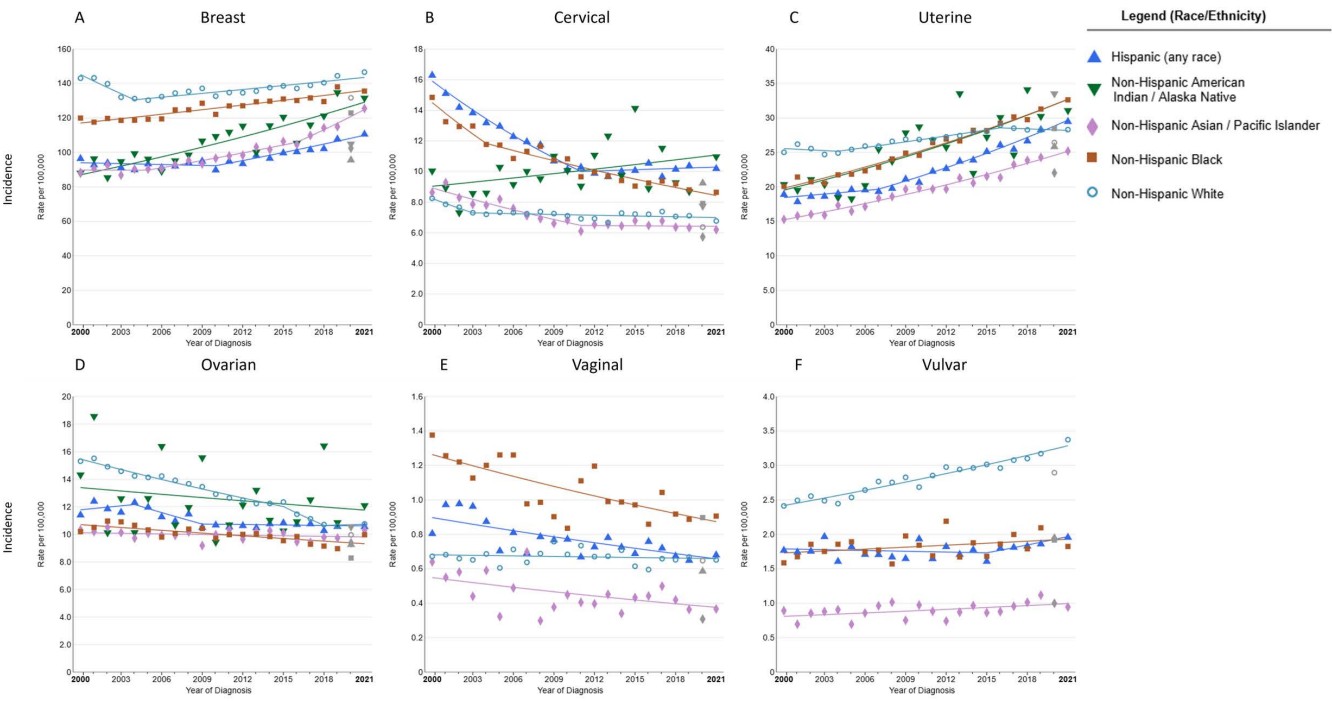

**Fig 1. Trends in annual age-standardized rates for six major female-specific cancers incidence (2000-2021) by race and ethnicity in the USA.** Trends in age-standardized incidence rates among (A) breast cancer, (B) cervical cancer, (C) uterine cancer, (D) ovarian cancer, (E) vaginal cancer, (F) vulvar cancer. Owing to sparse data, incidence for AIAN individuals of vaginal cancer and vulvar cancer are excluded.

The incidence (AAPC, −0.7; 95% CI, −1.9 to 0.2) and mortality (AAPC, −0.7; 95% CI, −0.8 to −0.5) rates of cervical cancer have generally shown a declining trend across all races from 2000 to 2021 (2000–2022) (Figs 1B and 2B). The incidence of cervical cancer decreased among Blacks, Whites, and AAPIs, but increased among AIANs and Hispanics (Fig 1B, Table 2). The mortality rates of cervical cancer decreased among all race/ethnicity groups, except for Whites, which remained stable (Fig 2B, Table 3). From 2000 to 2021, the incidence rates of uterine cancer have increased across all racial and ethnic groups, except for Whites, whose rates decreased in the early (from 2000 to 2004) and later years (from 2016

**Table 2. Joinpoint trends in age-standardized incidence rates for the major female-specific cancers by race and ethnicity in the United States, 2000–2021.**

| Cancer site | Trend 1 | | Trend 2 | | Trend 3 | | Trend 4 | | AAPC |
|---|---|---|---|---|---|---|---|---|---|
| | Years | APC | Years | APC | Years | APC | Years | APC | 2017–2021 |
| **Breast cancer** | | | | | | | | | |
| All Races | 2000–2004 | −2.3* | 2004–2016 | 0.4 | 2016–2021 | 1.4* | | | 1.4* (0.7 to 2.5) |
| White | 2000–2004 | −2.6* | 2004–2021 | 0.6* | | | | | 0.6* (0.3 to 0.9) |
| Black | 2000–2021 | 0.7* | | | | | | | 0.7* (0.6 to 0.9) |
| AIAN | 2000–2021 | 1.9* | | | | | | | 1.9* (1.5 to 2.4) |
| AAPI | 2000–2005 | 0 | 2005–2016 | 1.5* | 2016–2021 | 3.4* | | | 3.4* (2.4 to 4.7) |
| Hispanic | 2000–2010 | −0.2 | 2010–2021 | 1.6* | | | | | 1.6* (1.1 to 2.8) |
| **Cervical cancer** | | | | | | | | | |
| All Races | 2000–2002 | −5.0* | 2002–2013 | −1.5* | 2013–2016 | 1.8 | 2016–2021 | −0.7 | −0.7 (−1.9 to 0.2) |
| White | 2000–2003 | −3.9* | 2003–2021 | −0.2 | | | | | −0.2 (−0.5 to 0.2) |
| Black | 2000–2004 | −5.0* | 2004–2021 | −2.0* | | | | | −2.0* (−2.5 to −0.5) |
| AIAN | 2000–2021 | 1.0 | | | | | | | 1.0 (−0.3 to 2.4) |
| AAPI | 2000–2011 | −2.9* | 2011–2021 | −0.1 | | | | | −0.1 (−1.1 to 2.7) |
| Hispanic | 2000–2011 | −4.1 | 2011–2021 | 0.3 | | | | | 0.3 (−0.3 to 1.0) |
| **Uterine cancer** | | | | | | | | | |
| All Races | 2000–2003 | −0.8 | 2003–2021 | 1.3* | | | | | 1.3* (1.2 to 1.5) |
| White | 2000–2004 | −0.4 | 2004–2016 | 1.1* | 2016–2021 | −0.3 | | | −0.3 (−1.6 to 0.5) |
| Black | 2000–2021 | 2.4* | | | | | | | 2.4* (2.2 to 2.7) |
| AIAN | 2000–2021 | 2.5* | | | | | | | 2.5* (1.5 to 3.6) |
| AAPI | 2000–2021 | 2.4* | | | | | | | 2.4* (2.2 to 2.7) |
| Hispanic | 2000–2007 | 0.9 | 2007–2021 | 3.0* | | | | | 3.0* (2.6 to 4.0) |
| **Ovarian cancer** | | | | | | | | | |
| All Races | 2000–2021 | −1.5 | | | | | | | −1.5* (−1.7 to −1.4) |
| White | 2000–2015 | −1.7* | 2015–2018 | −4.0* | 2018–2021 | 0.2 | | | −0.8 (−2.6 to 0.3) |
| Black | 2000–2021 | −0.7* | | | | | | | −0.7* (−0.9 to −0.4) |
| AIAN | 2000–2021 | −0.6 | | | | | | | −0.6 (−2.2 to 1.1) |
| AAPI | 2000–2021 | −0.1 | | | | | | | −0.1 (−0.4 to 0.2) |
| Hispanic | 2000–2004 | 0.8 | 2004–2009 | −2.5 | 2009–2021 | −0.1 | | | −0.1 (−1.7 to 1.2) |
| **Vaginal cancer** | | | | | | | | | |
| All Races | 2000–2021 | −0.6* | | | | | | | −0.6* (−1.0 to −0.3) |
| White | 2000–2021 | −0.2 | | | | | | | −0.2 (−0.6 to 0.3) |
| Black | 2000–2021 | −1.7* | | | | | | | −1.7* (−2.6 to −0.7) |
| AIAN | 2000–2021 | — | | | | | | | — |
| AAPI | 2000–2021 | −1.8* | | | | | | | −1.8* (−3.1 to −0.3) |
| Hispanic | 2000–2021 | −1.5* | | | | | | | −1.5* (−2.1 to −0.8) |
| **Vulvar** | | | | | | | | | |
| All Races | 2000–2021 | 1.0* | | | | | | | 1.0* (0.8 to 1.1) |
| White | 2000–2021 | 1.5* | | | | | | | 1.5* (1.3 to 1.7) |
| Black | 2000–2021 | 0.5 | | | | | | | 0.5 (−0.3 to 1.3) |
| AIAN | 2000–2021 | 5.0* | | | | | | | 5.0* (1.6 to 9.7) |
| AAPI | 2000–2021 | 1.0* | | | | | | | 1.0* (0.1 to 2.0) |
| Hispanic | 2000–2015 | −0.2 | 2015–2021 | 2.2* | | | | | 2.2* (0 to 5.2) |

[a]All racial groups are exclusive of individuals identifying as Hispanic. Rates are per 100000 and age-adjusted to the 2000 US standard population. Estimates based on less than 16 cases are suppressed and not shown. Incidence data are from the National Cancer Institute Surveillance, Epidemiology and End Results database, and mortality data are from the National Center for Health Statistics database.

[b]Rates are adjusted for delays in case reporting.

[c]Rate ratios are based on rates age-adjusted to the 2000 US standard population, with White population as the reference group.

* Significant difference in rate ratio in comparison to the reference group ($p < 0.05$).

Abbreviation: AIAN = American Indian and Alaska Native; AAPI = Asian American and Pacific Islander.

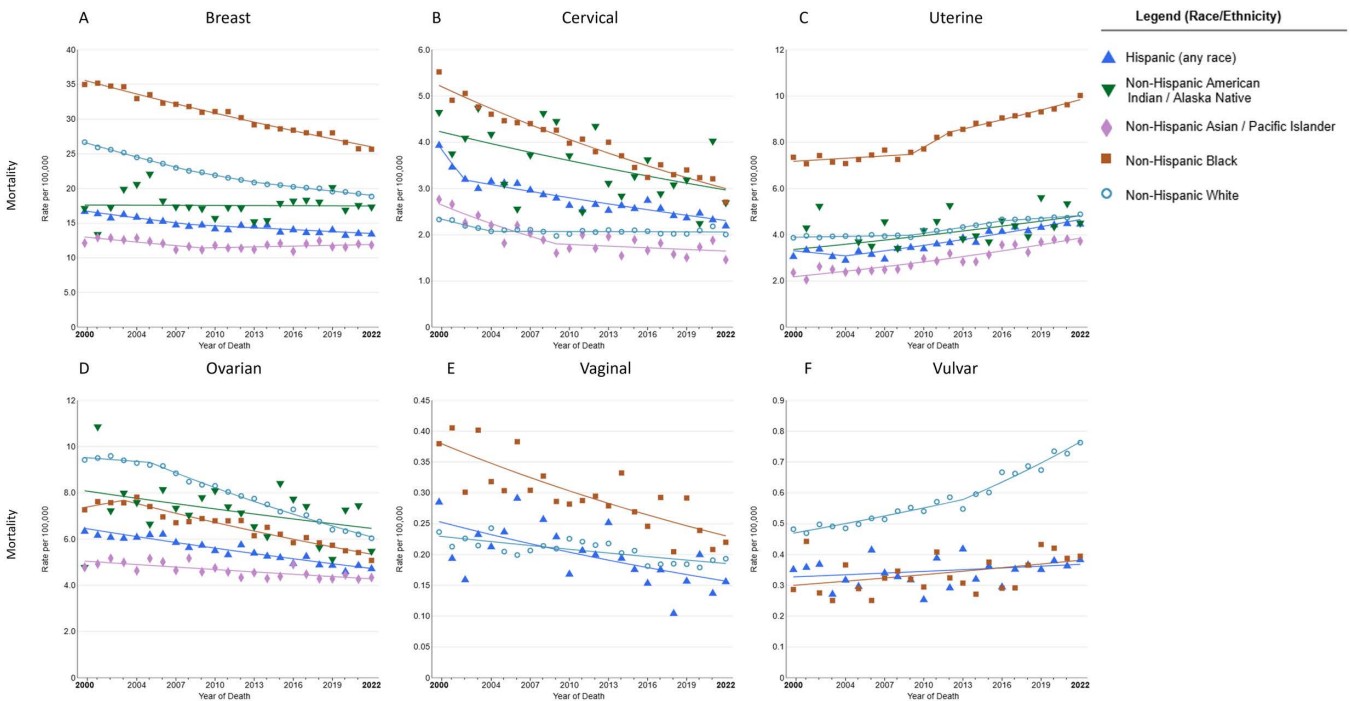

**Fig 2. Trends in annual age-standardized rates for six major female-specific cancers mortality (2000–2022) by race and ethnicity in the USA.** Trends in age-standardized mortality rates among (A) breast cancer, (B) cervical cancer, (C) uterine cancer, (D) ovarian cancer, (E) vaginal cancer, (F) vulvar cancer. Owing to sparse data, mortality for AAPI and AIAN individuals of vaginal cancer and vulvar cancer are excluded.

to 2021) (Fig 1C, Table 2). The mortality rates of uterine cancer have increased across all racial and ethnic groups (Fig 2C, Table 3). The incidence rates of ovarian cancer have declined across all ethnic groups, with particularly notable declines among Whites from 2000 to 2018, but remain stable from 2018 to 2021 (AAPC, 0.2) (Fig 1D, Table 2). The mortality rates of ovarian cancer have decreased across all ethnic groups, with the most significant decline among Whites (AAPC, −2.5; 95% CI, −2.7 to −2.3) (Fig 2D, Table 3). The incidence and mortality rates of vaginal cancer have decreased across all racial and ethnic groups in the most recent period (Figs 1E and 2E). Conversely, the incidence and mortality rates of vulvar cancer have increased across all racial and ethnic groups during the same period (Figs 1F and 2F).

### 3.3. Incidence and mortality by state

The incidence (2017–2021) and mortality rates (2018–2022) of FSCs show significant differences among Whites across 51 states in the United States (Figs 3 and 4, Table 4).

The District of Columbia had the highest breast cancer incidence rate (151.5; 95% CI, 141.1–162.5), while Utah had the lowest incidence rate (119; 95% CI, 116.2–121.7) (Fig 3A, Table 4). Hawaii had the highest breast cancer mortality rate (23.7; 95% CI, 20.6–27.2), whereas Massachusetts exhibited the lowest mortality rate (15.4; 95% CI, 14.8–16.0) (Fig 4A, Table 4). West Virginia (9.9; 95% CI, 9.0–11.0) had the highest incidence for cervical cancer, while the highest mortality rate of cervical cancer was observed in Oklahoma (3.6; 95% CI, 3.2–4.1) (Figs 3B and 4B, Table 4). The lowest incidence rate of cervical cancer was observed in the District of Columbia (4.2; 95% CI, 92.6–6.3), and the lowest mortality rate was found in Massachusetts (1.1; 95% CI, 0.9–1.3) (Figs 3B and 4B, Table 4). The highest incidence rate of uterine cancer was found in West Virginia (36.5; 95% CI, 34.9–38.2), while Alabama had the lowest incidence rate (19.7; 95% CI, 18.9–20.6) (Fig 3C, Table 4). Vermont (6.3; 95% CI, 5.3–7.4) had the highest mortality rate of uterine cancer and

**Table 3. Joinpoint trends in age-standardized mortality rates for the major female-specific cancers by race and ethnicity in the United States, 2000–2022.**

| Cancer site | Trend 1 | | Trend 2 | | Trend 3 | | Trend 4 | | AAPC |
|---|---|---|---|---|---|---|---|---|---|
| | Years | APC | Years | APC | Years | APC | Years | APC | 2018-2022 |
| **Breast cancer** | | | | | | | | | |
| All Races | 2000–2008 | −2.1* | 2008–2014 | −1.5* | 2014–2018 | −0.9* | 2018–2022 | −1.5* | −1.5* (−2.0 to −1.3) |
| White | 2000–2008 | −2.0* | 2008–2013 | −1.5* | 2013–2022 | −1.1* | | | −1.1* (−0.6 to −1.2) |
| Black | 2000–2022 | −1.4* | | | | | | | −1.4* (−1.5 to −1.3) |
| AIAN | 2000–2022 | 0 | | | | | | | 0 |
| AAPI | 2000–2009 | −1.4* | 2009–2022 | 0.3* | | | | | 0.3 (−0.1 to 2.1) |
| Hispanic | 2000–2008 | −1.5* | 2008–2022 | −0.7 | | | | | −0.7 (−0.9 to 0.8) |
| **Cervical cancer** | | | | | | | | | |
| All Races | 2000–2003 | −4.2* | 2003–2022 | −0.7* | | | | | −0.7* (−0.8 to −0.5) |
| White | 2000–2004 | −3.1* | 2004–2022 | 0 | | | | | 0 |
| Black | 2000–2022 | −2.5* | | | | | | | −2.5* (−2.8 to −2.2) |
| AIAN | 2000–2022 | −1.6* | | | | | | | −1.6* (−2.9 to −0.2) |
| AAPI | 2000–2009 | −4.3* | 2009–2022 | −0.7 | | | | | −0.7 (−2.5 to 7.8) |
| Hispanic | 2000–2002 | −9.9* | 2002–2020 | −1.6* | | | | | −1.6* (−2.9 to −0.2) |
| **Uterine cancer** | | | | | | | | | |
| All Races | 2000–2009 | 0.3 | 2009–2016 | 2.4* | 2016–2022 | 1.1 | | | 1.1* (0.3 to 1.5) |
| White | 2000–2009 | 0.2 | 2009–2016 | 2.1* | 2016–2022 | 0.8 | | | 0.8 (−0.4 to 1.4) |
| Black | 2000–2009 | 0.5 | 2009–2016 | 4.1* | 2016–2022 | 1.5* | | | 1.5* (0.6 to 1.9) |
| AIAN | 2000–2022 | 1.7* | | | | | | | 1.7* (0.2 to 3.5) |
| AAPI | 2000–2022 | 2.6* | | | | | | | 2.6* (2.1 to 3.4) |
| Hispanic | 2000–2004 | −1.7 | 2004–2022 | 2.3* | | | | | 2.3* (1.8 to 4.3) |
| **Ovarian cancer** | | | | | | | | | |
| All Races | 2000–2005 | −0.6 | 2005–2022 | −2.4* | | | | | −2.4* (−2.6 to −2.3) |
| White | 2000–2005 | −0.4 | 2005–2022 | −2.5* | | | | | −2.5* (−2.7 to −2.3) |
| Black | 2000–2003 | 1.4 | 2003–2022 | −1.9* | | | | | −1.9* (−3.1 to −1.6) |
| AIAN | 2000–2022 | −1.0 | | | | | | | −1.0 (−2.2 to 0.3) |
| AAPI | 2000–2022 | −0.7* | | | | | | | −0.7* (−1.1 to −0.4) |
| Hispanic | 2000–2022 | −1.4* | | | | | | | −1.4* (−1.6 to −1.2) |
| **Vaginal cancer** | | | | | | | | | |
| All Races | 2000–2022 | −1.2* | | | | | | | −1.2* (−1.6 to −0.9) |
| White | 2000–2022 | −1.0* | | | | | | | −1.0* (−1.5 to −0.5) |
| Black | 2000–2022 | −2.3* | | | | | | | −2.3* (−3.1 to −0.5) |
| AIAN | 2000–2021 | — | | | | | | | — |
| AAPI | 2000–2022 | −1.1 | | | | | | | −1.1 (−3.5 to 0.9) |
| Hispanic | 2000–2022 | −2.2* | | | | | | | −2.2* (−3.5 to −0.7) |
| **Vulvar cancer** | | | | | | | | | |
| All Races | 2000–2013 | 1.1 | 2013–2022 | 2.7* | | | | | 2.7* (2.0 to 4.5) |
| White | 2000–2013 | 1.6 | 2013–2022 | 3.2* | | | | | 3.2* (2.4 to 5.0) |
| Black | 2000–2022 | 1.1 | | | | | | | 1.1 (0 to 2.4) |
| AIAN | 2000–2022 | — | | | | | | | — |
| AAPI | 2000–2022 | 3.2* | | | | | | | 3.2* (1.3 to 6.2) |
| Hispanic | 2000–2022 | 0.5 | | | | | | | 0.5 (−0.2 to 1.5) |

[a]All racial groups are exclusive of individuals identifying as Hispanic. Rates are per 100000 and age-adjusted to the 2000 US standard population. Estimates based on less than 16 cases are suppressed and not shown. Incidence data are from the National Cancer Institute Surveillance, Epidemiology and End Results database, and mortality data are from the National Center for Health Statistics database.

[b]Rates are adjusted for delays in case reporting.

[c]Rate ratios are based on rates age-adjusted to the 2000 US standard population, with White population as the reference group.

*Significant difference in rate ratio in comparison to the reference group ($p < 0.05$).

Abbreviation: AIAN = American Indian and Alaska Native; AAPI = Asian American and Pacific Islander.

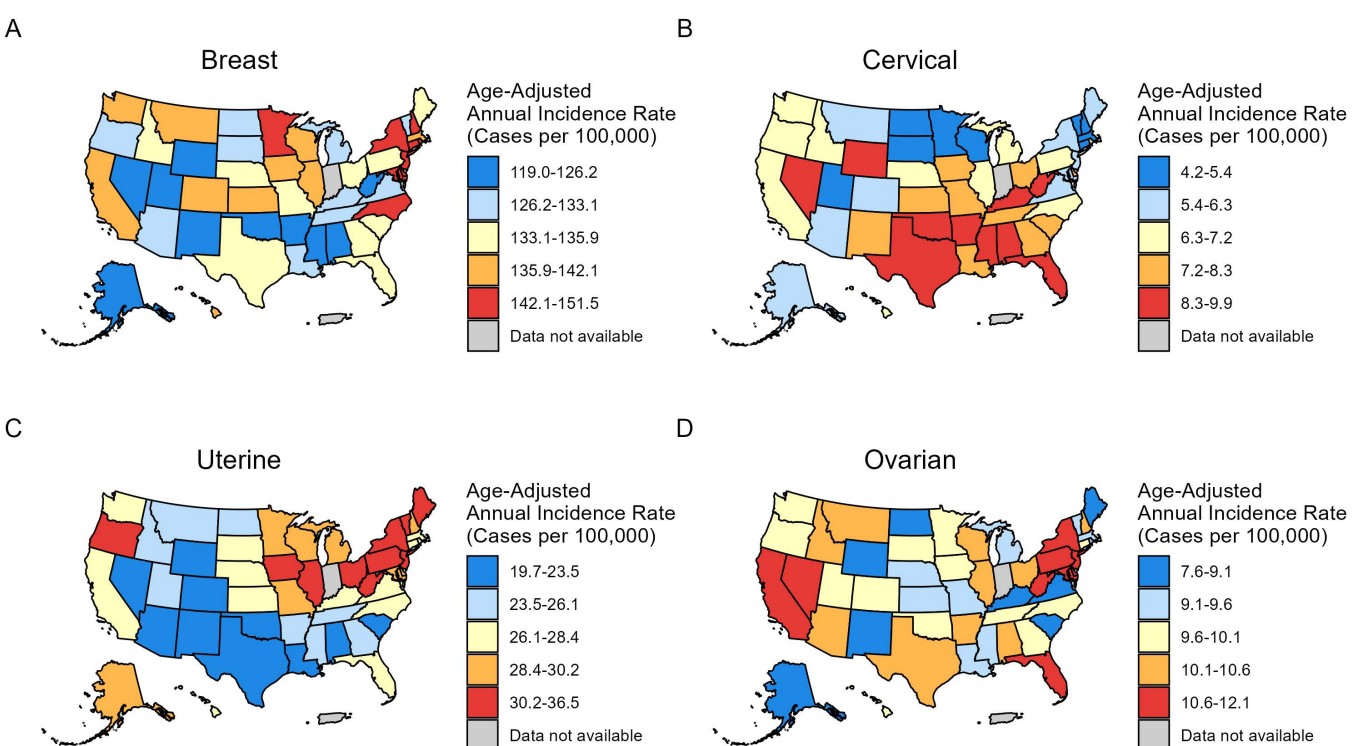

**Fig 3. Age-standardized four major female-specific cancers incidence (2017-2021) among non-Hispanic White population across 51 U.S. states.** (A) breast cancer, (B) cervical cancer, (C) uterine cancer, (D) ovarian cancer. Owing to sparse data, incidence of vaginal cancer and vulvar cancer are excluded.

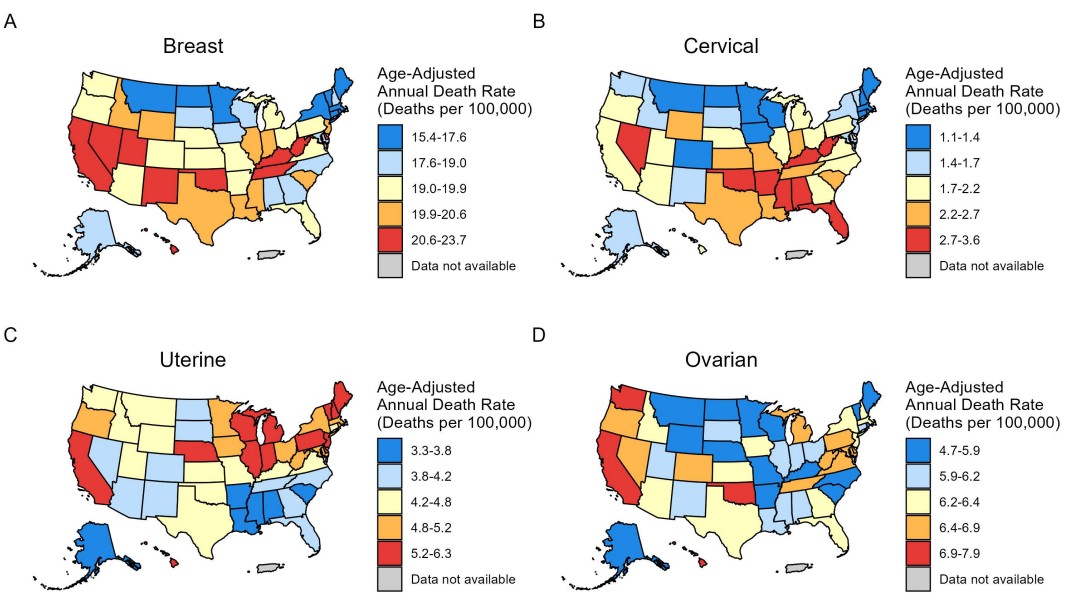

**Fig 4. Age-standardized four major female-specific cancers mortality (2018-2022) among non-Hispanic White population across 51 U.S. states.** (A) breast cancer, (B) cervical cancer, (C) uterine cancer, (D) ovarian cancer. Owing to sparse data, mortality of vaginal cancer and vulvar cancer are excluded.

**Table 4. Age-standardized incidence (2017–2021) and death (2018–2022) rate of four major female-related cancer among the non-Hispanic White population by state, United States.**

| Location | Incidence | | | | Mortality | | | |
|---|---|---|---|---|---|---|---|---|
| | Breast | Cervical | Uterine | Ovarian | Breast | Cervical | Uterine | Ovarian |
| Alabama | 122.3 (120.2-124.5) | 9.3 (8.6-10) | 19.7 (18.9-20.6) | 10.4 (9.8-11) | 18.6 (17.8-19.4) | 3.1 (2.8-3.5) | 3.3 (3-3.7) | 6.1 (5.7-6.6) |
| Alaska | 126.1 (119.7-132.7) | 6 (4.7-7.7) | 29.4 (26.5-32.6) | 8.3 (6.8-10.2) | 18.1 (15.8-20.7) | 1.7 (1-2.6) | 3.5 (2.5-4.7) | 5.9 (4.6-7.4) |
| Arizona | 127 (125.1-129) | 5.9 (5.4-6.4) | 23.5 (22.7-24.3) | 10.3 (9.7-10.8) | 19.6 (18.9-20.3) | 1.9 (1.6-2.2) | 3.9 (3.6-4.2) | 6.4 (6-6.9) |
| Arkansas | 123.6 (121-126.3) | 9.5 (8.7-10.4) | 25.1 (24-26.4) | 10.4 (9.6-11.2) | 19.3 (18.3-20.3) | 3.4 (2.9-3.9) | 3.8 (3.4-4.2) | 5.9 (5.4-6.5) |
| California | 140.1 (139-141.2) | 6.4 (6.1-6.6) | 27 (26.6-27.5) | 10.9 (10.6-11.2) | 21.3 (21-21.7) | 1.9 (1.8-2) | 5.3 (5.1-5.4) | 7.1 (6.9-7.4) |
| Colorado | 137.9 (135.8-140.1) | 5.6 (5.1-6) | 22.6 (21.8-23.4) | 9.7 (9.1-10.2) | 19.1 (18.4-19.9) | 1.4 (1.2-1.7) | 4.1 (3.8-4.5) | 6.5 (6.1-7) |
| Connecticut | 148.1 (145.4-150.9) | 4.7 (4.1-5.2) | 28.4 (27.3-29.5) | 9.8 (9.1-10.5) | 17.2 (16.3-18) | 1.4 (1.2-1.8) | 4.4 (4-4.8) | 6.1 (5.7-6.7) |
| Delaware | 143.4 (138.1-148.8) | 7.4 (6.1-8.9) | 29.3 (27.1-31.6) | 10.7 (9.3-12.2) | 21.5 (19.6-23.4) | 2 (1.4-2.8) | 5.3 (4.4-6.3) | 6.6 (5.7-7.7) |
| District of Columbia | 151.5 (141.1-162.5) | 4.2 (2.6-6.3) | 26.3 (22.1-31.1) | 11.4 (8.7-14.8) | 15.6 (12.5-19.2) | data not available | 4.8 (3.2-7) | 7.6 (5.5-10.4) |
| Florida | 134.3 (133.1-135.5) | 9 (8.6-9.4) | 26.7 (26.2-27.2) | 11.6 (11.2-11.9) | 19.1 (18.7-19.5) | 2.8 (2.6-3) | 4.1 (3.9-4.3) | 6.4 (6.1-6.6) |
| Georgia | 134.7 (133-136.4) | 7.9 (7.4-8.4) | 24.6 (23.9-25.3) | 10 (9.6-10.5) | 19 (18.4-19.6) | 2.2 (1.9-2.4) | 3.9 (3.7-4.2) | 6.4 (6.1-6.8) |
| Hawaii | 140.2 (132.1-148.7) | 7.2 (5.3-9.5) | 27 (23.9-30.5) | 10.1 (8-12.7) | 23.7 (20.6-27.2) | 2.2 (1.2-3.8) | 5.3 (4-7) | 7.9 (6.3-10) |
| Idaho | 134.6 (131.2-138.2) | 6.5 (5.7-7.4) | 25 (23.6-26.5) | 10.3 (9.4-11.3) | 20.4 (19.2-21.8) | 1.7 (1.3-2.1) | 4.3 (3.7-4.9) | 6.3 (5.7-7.1) |
| Illinois | 139.8 (138.3-141.3) | 6.7 (6.3-7) | 30.4 (29.8-31.1) | 10.5 (10.1-10.9) | 20 (19.4-20.5) | 2.1 (1.9-2.3) | 5.4 (5.1-5.6) | 6.2 (5.9-6.5) |
| Indiana | data not available | | | | 20.3 (19.7-21) | 2.7 (2.5-3) | 5.3 (4.9-5.6) | 6.1 (5.8-6.5) |
| Iowa | 139.1 (136.5-141.7) | 7.5 (6.8-8.2) | 30.7 (29.6-31.9) | 9.7 (9.1-10.4) | 17.9 (17.1-18.8) | 1.4 (1.2-1.7) | 5.1 (4.7-5.6) | 6.3 (5.8-6.8) |
| Kansas | 137.2 (134.4-140.1) | 7.8 (7-8.6) | 26.9 (25.7-28.1) | 9.2 (8.4-9.9) | 19.9 (18.9-20.9) | 2.6 (2.2-3.1) | 4.7 (4.2-5.2) | 6.4 (5.8-7) |
| Kentucky | 130 (127.9-132.1) | 9.8 (9.1-10.4) | 28.1 (27.2-29.1) | 8.8 (8.3-9.4) | 21.3 (20.5-22.1) | 3 (2.6-3.3) | 4.4 (4.1-4.8) | 5.6 (5.2-6) |
| Louisiana | 130.2 (127.8-132.7) | 8.2 (7.5-8.9) | 20.6 (19.6-21.6) | 9.2 (8.5-9.9) | 20 (19.1-21) | 2.3 (2-2.7) | 3.5 (3.1-3.9) | 6 (5.6-6.5) |
| Maine | 134.2 (130.7-137.8) | 6.1 (5.3-7.1) | 31.9 (30.3-33.6) | 8.4 (7.5-9.4) | 16.8 (15.6-18) | 1.3 (0.9-1.7) | 5.5 (4.9-6.1) | 5.8 (5.2-6.5) |
| Maryland | 144.4 (142.1-146.9) | 6.1 (5.5-6.6) | 28.5 (27.5-29.5) | 10.8 (10.2-11.4) | 18.6 (17.9-19.5) | 1.7 (1.5-2) | 5 (4.6-5.4) | 6.6 (6.2-7.1) |
| Massachusetts | 142.1 (140.2-143.9) | 4.3 (3.9-4.6) | 28.4 (27.6-29.2) | 9.6 (9.2-10.1) | 15.4 (14.8-16) | 1.1 (0.9-1.3) | 5.1 (4.8-5.4) | 6.4 (6.1-6.8) |
| Michigan | 129.6 (128.1-131.1) | 6.4 (6-6.8) | 29.3 (28.6-30) | 9.6 (9.2-10) | 19.7 (19.1-20.2) | 1.9 (1.7-2.1) | 5.4 (5.1-5.7) | 6.6 (6.3-6.9) |
| Minnesota | 143.6 (141.6-145.6) | 4.8 (4.3-5.2) | 30.2 (29.4-31.1) | 9.7 (9.2-10.2) | 17.3 (16.6-17.9) | 1.2 (1-1.4) | 5 (4.6-5.3) | 5.9 (5.5-6.3) |
| Mississippi | 124 (121-127) | 9.5 (8.5-10.5) | 23.6 (22.3-24.9) | 9.4 (8.6-10.3) | 20.1 (19-21.3) | 3.3 (2.8-3.8) | 3.6 (3.2-4.2) | 6 (5.4-6.7) |
| Missouri | 134.8 (133-136.7) | 8.3 (7.8-8.9) | 28.7 (27.8-29.5) | 9.3 (8.8-9.8) | 19.3 (18.6-20) | 2.4 (2.1-2.7) | 4.8 (4.5-5.2) | 5.8 (5.4-6.1) |
| Montana | 137 (132.6-141.4) | 6.3 (5.3-7.4) | 25.2 (23.5-27.1) | 10.2 (9-11.5) | 17.3 (15.9-18.7) | 1.2 (0.8-1.7) | 4.5 (3.8-5.3) | 5.8 (5.1-6.7) |
| Nebraska | 134.6 (131.3-138.1) | 7.2 (6.4-8.2) | 28 (26.5-29.5) | 9.2 (8.4-10.2) | 19.8 (18.6-21.1) | 2 (1.6-2.5) | 5.3 (4.7-6) | 5.7 (5.1-6.4) |
| Nevada | 121 (118-124.2) | 8.7 (7.7-9.7) | 23.1 (21.8-24.5) | 11.2 (10.3-12.3) | 23.4 (22.1-24.8) | 2.8 (2.4-3.4) | 4.2 (3.7-4.8) | 6.9 (6.2-7.6) |
| New Hampshire | 142.4 (138.6-146.2) | 5 (4.2-5.8) | 30 (28.4-31.7) | 10.3 (9.3-11.4) | 17.9 (16.6-19.2) | 1.3 (1-1.8) | 5.5 (4.8-6.2) | 6.4 (5.7-7.2) |
| New Jersey | 148.5 (146.6-150.4) | 6.3 (5.9-6.8) | 33 (32.1-33.8) | 11.4 (10.9-11.9) | 20.2 (19.6-20.9) | 1.7 (1.5-1.9) | 5.7 (5.4-6) | 6.3 (6-6.7) |
| New Mexico | 125 (120.7-129.5) | 7.3 (6.1-8.8) | 23 (21.2-24.8) | 9.1 (8-10.4) | 21.7 (20.1-23.5) | 1.7 (1.2-2.4) | 4.2 (3.5-5) | 6.1 (5.2-7) |
| New York | 144.4 (143.1-145.7) | 6 (5.8-6.3) | 31.5 (31-32.1) | 11.5 (11.2-11.9) | 17.6 (17.2-18) | 1.5 (1.4-1.7) | 5.2 (5-5.4) | 6.4 (6.2-6.7) |
| North Carolina | 145.8 (144.2-147.5) | 6.6 (6.2-7) | 26.3 (25.6-26.9) | 9.7 (9.3-10.1) | 18.8 (18.3-19.4) | 1.8 (1.6-2) | 4.2 (3.9-4.4) | 5.8 (5.5-6.1) |
| North Dakota | 131.9 (126.6-137.4) | 4.9 (3.8-6.1) | 26.1 (23.9-28.6) | 7.6 (6.4-9) | 16.3 (14.5-18.2) | 1.2 (0.7-1.8) | 4.1 (3.3-5.1) | 4.7 (3.8-5.7) |
| Ohio | 134.4 (133-135.7) | 8 (7.6-8.3) | 31.9 (31.3-32.5) | 10.4 (10-10.8) | 19.8 (19.3-20.3) | 2.2 (2-2.4) | 5.1 (4.9-5.4) | 6.1 (5.8-6.3) |
| Oklahoma | 123.3 (120.9-125.7) | 9.3 (8.6-10.1) | 23.5 (22.5-24.6) | 10.1 (9.4-10.8) | 22.6 (21.6-23.6) | 3.6 (3.2-4.1) | 4.4 (4-4.8) | 7 (6.5-7.6) |
| Oregon | 133.1 (130.9-135.3) | 6.5 (6-7.1) | 30.3 (29.3-31.4) | 9.9 (9.3-10.5) | 19.9 (19.1-20.7) | 1.9 (1.6-2.2) | 5.2 (4.8-5.6) | 6.8 (6.4-7.3) |
| Pennsylvania | 133.9 (132.6-135.2) | 6.6 (6.2-6.9) | 32.7 (32.1-33.3) | 10.9 (10.6-11.3) | 19.2 (18.8-19.7) | 1.8 (1.7-2) | 5.5 (5.3-5.7) | 6.5 (6.2-6.7) |
| Puerto Rico | only available for All Races | | | | | | | |
| Rhode Island | 142.6 (138-147.3) | 6.7 (5.6-8.1) | 28.4 (26.5-30.4) | 9.2 (8.1-10.5) | 16.7 (15.3-18.2) | 1.3 (0.8-1.8) | 4.9 (4.2-5.8) | 5.5 (4.8-6.5) |
| South Carolina | 135.9 (133.6-138.2) | 7.9 (7.3-8.6) | 22.7 (21.8-23.6) | 8.7 (8.1-9.3) | 20.2 (19.4-21) | 2.3 (2-2.7) | 3.8 (3.5-4.2) | 5.3 (4.9-5.8) |
| South Dakota | 132.9 (127.9-138) | 5.4 (4.3-6.6) | 26.6 (24.5-28.9) | 10.1 (8.8-11.6) | 18 (16.4-19.9) | 1.6 (1.1-2.3) | 3.9 (3.2-4.8) | 6.1 (5.1-7.1) |

*(Continued)*

**Table 4.** (Continued)

| Location | Incidence | | | | Mortality | | | |
|---|---|---|---|---|---|---|---|---|
| | Breast | Cervical | Uterine | Ovarian | Breast | Cervical | Uterine | Ovarian |
| Tennessee | 126.4 (124.7-128.2) | 7.7 (7.2-8.2) | 25.5 (24.8-26.3) | 10 (9.5-10.5) | 20.9 (20.3-21.6) | 2.5 (2.2-2.7) | 4.1 (3.8-4.4) | 6.5 (6.1-6.8) |
| Texas | 133.4 (132.2-134.6) | 8.8 (8.5-9.1) | 23.3 (22.9-23.8) | 10.6 (10.2-10.9) | 20.6 (20.2-21.1) | 2.7 (2.6-2.9) | 4.4 (4.2-4.6) | 6.4 (6.2-6.7) |
| Utah | 119 (116.2-121.7) | 5.3 (4.7-5.9) | 25.6 (24.4-26.9) | 9.7 (8.9-10.5) | 20.7 (19.6-21.9) | 1.8 (1.4-2.1) | 4.6 (4.1-5.1) | 6.2 (5.6-6.8) |
| Vermont | 127.6 (122.5-132.9) | 5.4 (4.3-6.8) | 31.6 (29.3-34.1) | 9.3 (8-10.9) | 17.1 (15.4-19) | 1.6 (1.1-2.4) | 6.3 (5.3-7.4) | 5.9 (5-7.1) |
| Virginia | 131.9 (130.1-133.7) | 5.5 (5.1-5.9) | 26.3 (25.6-27.1) | 9 (8.6-9.5) | 19.5 (18.9-20.1) | 1.8 (1.6-2) | 4.6 (4.3-4.9) | 6.6 (6.3-7) |
| Washington | 140.9 (139.1-142.7) | 6.4 (5.9-6.8) | 26.8 (26-27.5) | 9.7 (9.2-10.2) | 19.6 (19-20.3) | 1.7 (1.5-1.9) | 4.8 (4.5-5.2) | 7.1 (6.7-7.5) |
| West Virginia | 126 (123-129.1) | 9.9 [9–11] | 36.5 (34.9-38.2) | 12.1 (11.1-13.1) | 21.2 (20-22.4) | 3.1 (2.6-3.7) | 5 (4.5-5.6) | 6.7 (6-7.4) |
| Wisconsin | 138.5 (136.5-140.4) | 5.4 (5-5.8) | 30.2 (29.4-31.1) | 10.4 (9.9-11) | 17.8 (17.1-18.4) | 1.3 (1.1-1.5) | 5.3 (5-5.7) | 5.9 (5.6-6.3) |
| Wyoming | 126.2 (120.3-132.3) | 8.4 (6.8-10.3) | 22.9 (20.5-25.4) | 8.6 (7.2-10.3) | 20.3 (18.1-22.7) | 2.3 (1.5-3.3) | 4.4 (3.4-5.5) | 5.9 (4.7-7.2) |

the lowest mortality rate was found in Alabama (3.3; 95% CI, 3.0–3.7) (Fig 4C, Table 4). For ovarian cancer, West Virginia (12.1; 95% CI, 11.1–13.1) had the highest incidence rate, while Hawaii (7.9; 95% CI, 6.3–10.0) had the highest mortality rate. North Dakota had the lowest incidence (7.6; 95% CI, 6.4–9.0) and mortality (4.7; 95% CI, 3.8–5.7) rates (Figs 3D and 4D, Table 4).

### 3.4. Risk-attributable deaths over time

The results of four major FSCs attributable to the risk factors are shown in Fig 5.

From 2000 to 2021, the deaths rates for all breast cancer decreased. Dietary risks contributed to deaths of breast cancer decreased by 26.8%, followed by reductions in risks from high body mass index, high fasting plasma glucose, high alcohol use, tobacco use and low physical activity (Fig 5A). The deaths of cervical cancer caused by risk factors were decreased from 2000 to 2021, with unsafe sex decreasing by 19.8% and tobacco use decreasing by 31.5% (Fig 5B). In contrast, for uterine cancer, the deaths attributable to high body mass index increased by 38.5% (Fig 5C). For ovarian cancer, the deaths attributable to high body mass index decreased by 17.7% and occupational risks decreased by 30.3% (Fig 5D).

### 3.5. Risk-attributable deaths by state

The primary risk factor led to breast cancer in the US in 2000 was dietary risks, accounting for 35.9% of deaths, which showed a declining trend to 31.8% in 2021 (Fig 6A).

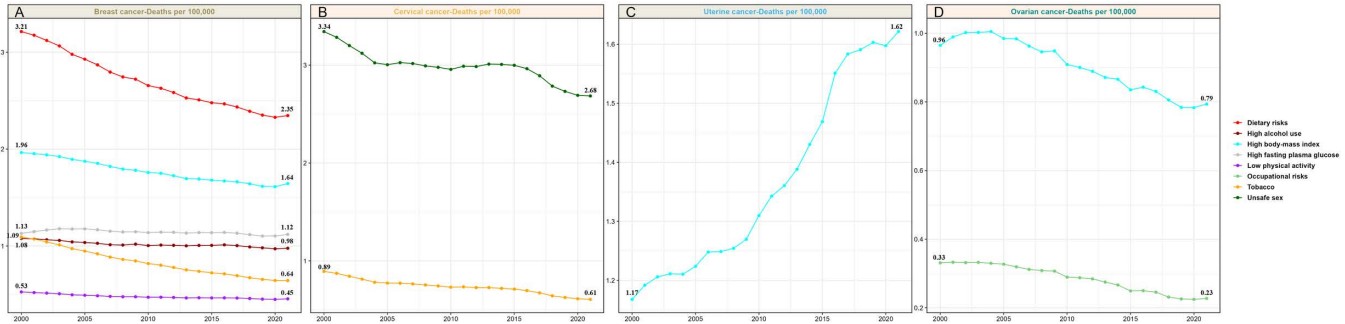

**Fig 5. Trends in age-standardized death rates attributable to specific risk factors for four major female-specific cancers incidence (2000–2021) in the USA.** (A) breast cancer, (B) cervical cancer, (C) uterine cancer, (D) ovarian cancer. Owing to sparse data of GBD database, death rate of vaginal cancer and vulvar cancer are excluded.

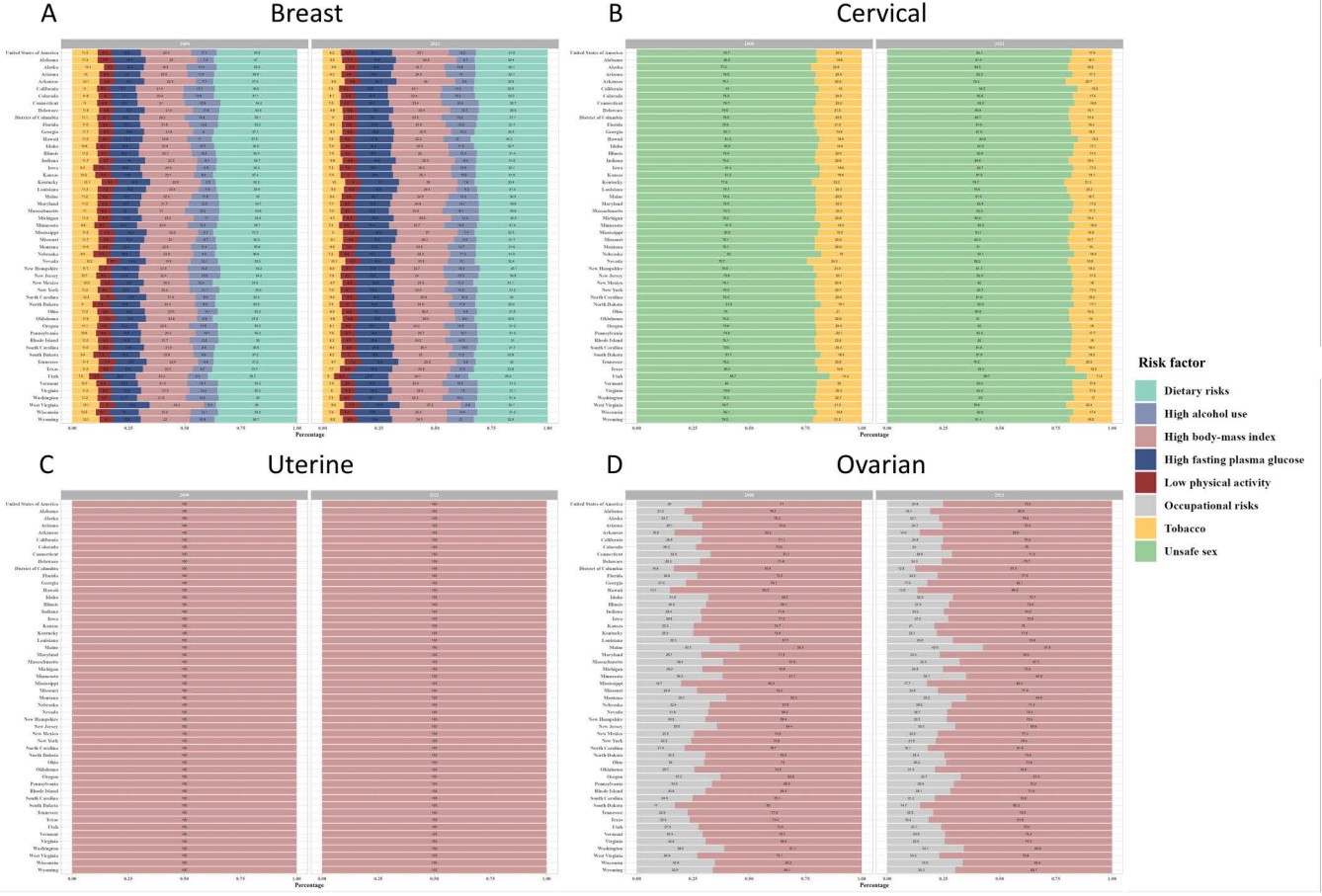

**Fig 6. Distribution of deaths caused by risk factors for four major female-specific cancers between 2000 and 2021 across 51 U.S. states.** (A) breast cancer, (B) cervical cancer, (C) uterine cancer, (D) ovarian cancer. Owing to sparse data, deaths of vaginal cancer and vulvar cancer are excluded.

Subsequent risk factors included high body mass index, high alcohol consumption, tobacco use, high fasting plasma glucose, and low physical activity. The contribution of dietary risks to breast cancer was notably heterogeneous across US states, with Utah showing the highest percentage (39.7% in 2000 and 35.2% in 2021) and New Hampshire the lowest (34.2% in 2000 and 30.1% in 2021) (Fig 6A). For cervical cancer, Utah had the highest percentage of cervical cancer deaths attributable to unsafe sex (85.7% in 2000 and 88.7% in 2021). Nevada had the lowest percentage in 2000 (75.7%), but Kentucky had the lowest percentage in 2021 (80.1%) (Fig 6B). For ovarian cancer, Hawaii had the highest percentage of ovarian cancer deaths attributable to high body-mass index in 2000 (87.5%), but District of Columbia had the highest percentage in 2021 (90.3%) (Fig 6C). Maine had the lowest percentage of ovarian cancer deaths attributable to high body-mass index (54.3% in 2000 and 57.4% in 2021) (Fig 6D).

## 4. Discussion

This study utilized high-quality, nationally representative data spanning over two decades, enabling a comprehensive analysis of temporal trends in incidence and mortality for six major female-specific cancers (FSCs) in the US. FSCs account for nearly one-third to one-half of all newly diagnosed cancers in women annually. These malignant tumors pose

a significant threat to women's health and well-being, especially profoundly affecting reproductive health [2]. We also estimated the burden of four major FSCs attributable to risk factors using GBD 2021 data.

Our findings confirm that breast cancer incidence remains highest among White women, while Black women experience disproportionately higher mortality—a pattern consistent with previous study [16]. These disparities may be attributed to multiple factors, including socioeconomic status, cultural beliefs, and social inequities [17]. Women with lower socioeconomic status face barriers to screening rates, later-stage diagnosis, and disparate treatments quality, contributing to poorer outcomes [18]. Compared to White women, Black women are more likely to delay seeking immediate treatment for breast symptoms [19] and experience higher rates of aggressive subtypes like triple-negative breast cancer [20]. Regional variations in risk factors, such as alcohol consumption, may also contribute to geographic disparities [21]. Rising breast cancer incidence is associated with increasing obesity, physical inactivity, and alcohol use [22,23]. Notably, while deaths from all breast cancer risk factors declined from 2000–2021, potentially reflecting improved screening and therapies [24], the proportional contribution of metabolic and behavioral risks increased.

Cervical cancer incidence decreased among most racial groups but increased among non-Hispanic American Indian and Alaska Native (AIAN) women, who also experience lower screening rates due to structural barriers such as geographic isolation, under-resourced healthcare, and socioeconomic challenges [25]. Consistently, compared to other racial and ethnic groups, the mortality rate of cervical cancer among Black women was higher. A recent study indicated that the prevalence of HPV is highest among Black women, and high-risk HPV infections persist longer in this population [26]. This may serve as a biological basis for the higher incidence of cervical cancer among Black women. Our research shows that from 2000 to 2021, the deaths of cervical cancer caused by risk factors were decreased. The observed decline in cervical cancer deaths underscores the importance of HPV vaccination, screening, and precancer treatment [27], alongside reduced smoking [28]. However, it is unclear why cervical cancer incidence and mortality among Whites is highest in the South region, including West Virginia. Further research is warranted to elucidate the underlying reasons for these regional disparities.

The incidence and mortality rates of uterine cancer have been increasing across nearly all racial groups, underscoring the need for enhanced research and prevention strategies. Black women had twice the mortality of other groups, possibly due to later-stage diagnosis and more aggressive tumor types [29]. The incidence rate of uterine cancer is highest in the Northeast and Midwest regions [30], consistent with our research findings. The high uterine cancer incidence in West Virginia may reflect both obesity prevalence and limited access to minimally invasive hysterectomy [30]. State-level Medicaid expansion policies represent a potential lever for intervention. Critically, deaths attributable to high body-mass index rose steadily over 21 years, reinforcing a strong epidemiological association between obesity and uterine cancer [31]. This suggests weight management interventions could be impactful mitigation strategies, such as bariatric surgery [32].

Over the past two decades, the incidence and mortality rates of ovarian cancer have decreased across nearly all racial groups, likely due to the increased acceptance and widespread use of oral contraceptives [33,34]. This significant progress can also be attributed to advancements in precise treatment methods for ovarian cancer [35]. An increasing body of evidence suggests that obesity is associated with a higher risk of ovarian cancer [36]. Deaths due to obesity-related risk factors have declined from 2000 to 2021, indicating that people have made significant progress in weight management in the United States.

Over the past two decades, both the incidence and mortality rates of vaginal cancer have declined across all racial groups. While HPV vaccination holds promise for prevention [37], its efficacy across age groups needs evaluation, especially as vaginal cancer primarily affects older women [38]. Persistently higher rates among Black and Hispanic women may stem from delayed diagnosis due to structural barriers to healthcare access, such as insurance coverage and regular care, leading to progression to invasive disease [38]. Factors associated with the risk of vaginal cancer include low socioeconomic status, multiple sexual partners, a family history of anogenital cancers, marital status, and smoking [39]. Although smoking is associated with vaginal cancer, the mechanism by which smoking induces the development of vaginal cancer remains unclear. Previous studies suggest that nicotine may block apoptosis and suppress the immune system, potentially promoting the development of vaginal cancer [38].

The incidence of vulvar cancer is highest among white women, potentially reflecting disparities in HPV prevalence [40]. Notably, although both vaginal and vulvar cancers are associated with HPV, the incidence and mortality rates of vulvar cancer have been rising across all racial groups over the past 20 years, consistent with previous research findings [41]. One possible explanation could be attributed to the latency period and delays in diagnosis [42]; the exact reasons for these divergent trends require further study.

This study has several limitations. First, sparse state-level data prevented geographic analysis for non-White groups. Second, potential racial/ethnic misclassification in records may underestimate rates for non-White/Black groups. Third, while GBD 2021 provides robust risk-attributed burden estimates, its lack of racial stratification precluded analysis of disparities in modifiable risks—a critical need highlighted by our SEER findings. Finally, the absence of data on vaginal and vulvar cancers in the GBD database precludes further investigation into the risk factors for these two cancer types.

## 5. Conclusion

Using high-quality population-based data, our study revealed a strong heterogeneity in incidence/mortality trends for six female-specific cancers (FSCs) across the US. Further efforts to prevent FSCs should focus on the management of risk factors and intervention measures, including expanding HPV vaccination in underserved communities, implementing Medicaid coverage expansion to improve treatment access, and developing targeted obesity prevention initiatives in high-risk regions. Further research should elucidate these patterns and mitigate disparities, particularly studies examining structural barriers in AIAN populations, state-level policy impacts on uterine cancer, and vaginal/vulvar cancer risk factors in vulnerable groups. Meanwhile, culturally tailored navigation programs and community-based screening can reduce racial/ethnic mortality disparities through improved care access. A comprehensive long-term approach requires obesity prevention and continuous registry monitoring to evaluate progress toward equity goals.

## Acknowledgments

The authors would like to thank the SEER, NCHS, and GBD databases and the reviewers for their professional advice.

## Author contributions

**Conceptualization:** Fan Li, Ze Zhang, Yunhai Li, Hongbo Huang, Tingting Wei, Ying Huang, Xiuquan Qu, Aijie Zhang, Yijing Xu, Jiaying Li, Zheng Gong, Zhiqi Hu.

**Data curation:** Fan Li, Ze Zhang, Yunhai Li, Hongbo Huang, Tingting Wei, Ying Huang, Xiuquan Qu, Aijie Zhang, Yijing Xu, Jiaying Li, Zheng Gong, Zhiqi Hu.

**Methodology:** Ze Zhang, Hongbo Huang, Tingting Wei.

**Resources:** Fan Li, Yunhai Li.

**Software:** Ze Zhang.

**Supervision:** Fan Li, Yunhai Li.

**Visualization:** Ze Zhang.

**Writing – original draft:** Ze Zhang.

**Writing – review & editing:** Fan Li, Yunhai Li.

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
