## [Decision Letter · Decision Letter 0]

17 Jul 2025

PONE-D-25-31996Disparities and Trends of the Incidence and Mortality of Female-Specific Cancers in the United StatesPLOS ONE

Dear Dr. Li,

Thank you for submitting your manuscript to PLOS ONE. After careful consideration, we feel that it has merit but does not fully meet PLOS ONE’s publication criteria as it currently stands. Therefore, we invite you to submit a revised version of the manuscript that addresses the points raised during the review process.

We look forward to receiving your revised manuscript.

Kind regards,

Kehinde S. Okunade

Academic Editor

PLOS ONE

-DOI: 10.14309/ajg.0000000000003198

-https://doi.org/10.1016/j.eururo.2022.11.023

In your revision ensure you cite all your sources (including your own works), and quote or rephrase any duplicated text outside the methods section. Further consideration is dependent on these concerns being addressed.

 [This study was supported by the National Natural Science Foundation of China (grant 82202913 and 82372996) and the Natural Science Foundation of Chongqing (grant CSTB 2023 NSCQ-MSX0480).]. 

4. In the online submission form, you indicated that [The data utilized in this study can be found in the Surveillance, Epidemiology, and End Results database Public-access SEER data (https://seer.cancer.gov/), National Center for Health Statistics (NCHS), and 2021 Global Burden of Disease (GBD) study (https://vizhub.healthdata.org/gbd-results/). Analytic code and any additional information required to reanalyze the data reported in this paper can be obtained from the corresponding author upon reasonable request, following the publication of this article.].

6. Please include a separate caption for each figure in your manuscript.

Additional Editor Comments (if provided):

Reviewers' comments:

Reviewer's Responses to Questions

**Comments to the Author**

1. Is the manuscript technically sound, and do the data support the conclusions?

Reviewer #1: Partly

Reviewer #2: Yes

2. Has the statistical analysis been performed appropriately and rigorously? 

Reviewer #1: Yes

Reviewer #2: Yes

3. Have the authors made all data underlying the findings in their manuscript fully available?

Reviewer #1: Yes

Reviewer #2: Yes

4. Is the manuscript presented in an intelligible fashion and written in standard English?

Reviewer #1: No

Reviewer #2: Yes

5. Review Comments to the Author

Reviewer #1: General Comments

This manuscript addresses a critically important topic concerning the racial, ethnic, and geographic disparities in incidence and mortality trends of six female-specific cancers in the United States. The authors utilize robust national databases (SEER and GBD) and appropriate statistical methods (Joinpoint regression, APC/AAPC) to evaluate temporal trends. The study is well-suited to inform health policy and cancer prevention strategies.

However, the manuscript requires substantial revisions to improve clarity, scientific rigor, and structure. Issues related to writing quality, data interpretation, and discussion of health disparities must be addressed before the paper is suitable for publication.

Specific Comments

1. Title and Abstract

•The abstract is dense with data and could overwhelm readers. Please prioritize key findings and move granular statistics to the results section.

•Clarify the study’s objective explicitly in the abstract.

•Language issues, such as “impacts on healthcare and economy,” should be revised for clarity (e.g., "healthcare systems and economic burden").

2. Introduction

•The background is informative but includes some redundancy. Streamline citations and focus on how this study fills gaps in existing literature. Also, use more current references.

•Define “disadvantaged populations” and consider referencing social determinants of health more explicitly to ground the disparities framework.

3. Methods

•Clarify statistical methods such as the Tiwari method and Joinpoint regression in plain terms for non-specialist readers.

•Ensure that all data sources and coding schemes (ICD-10, ICD-O-3) are clearly defined.

•The rationale for examining both long and short-term trends should be better justified.

4. Results

•The results section is overly technical and could benefit from subheadings and clearer narrative transitions (e.g., "Incidence by Race", "Mortality by State").

•Ensure all acronyms (e.g., AAPI, AIAN, AAPC) are defined at first use in each section.

•Avoid inconsistent reporting formats (e.g., confidence intervals vs. brackets vs. parentheses).

•Include visuals, such as maps or tables, referenced in the narrative for improved readability.

5. Discussion

•Expand the discussion on the root causes of disparities, including systemic inequities, access to screening and treatment, and socioeconomic factors.

•Avoid overstatements or causal implications. For instance, “urgent need” can be softened to “highlight the importance of...”

•Acknowledge the strengths of the study (e.g., large datasets, national scope).

6. Conclusion

•Strengthen the conclusion by offering specific recommendations or implications for policy, practice, or future research (e.g., expand HPV vaccination, improve Medicaid coverage).

•Emphasize the importance of equity-driven cancer control efforts.

7. Language and Style

•The manuscript would benefit from professional English editing to correct awkward phrasing, verb tense inconsistencies, and repetitive language.

•Suggested revision examples:

“Raised among AAPIs” → “Increased among Asian American and Pacific Islander populations”

“Had higher burdens” → “Experienced higher incidence and mortality rates”

Minor Issues

•Ensure that all figures and tables referenced (e.g., Table 4, Figures 1–4) are included and clearly labeled.

•Abbreviations should be consistent throughout (e.g., “FSCs” vs. “female-specific cancers”).

•Clarify any suppressed data points and include a rationale (e.g., <16 cases suppressed per SEER guidelines).

Summary

This study contributes significantly to the literature on cancer epidemiology and racial disparities in the U.S., but requires revision for clarity, completeness, and scientific precision. I recommend revision.

Reviewer #2: I've reviewed your manuscript, and I'm happy to report that it looks great! Here are some specific things that caught my attention:

- Your manuscript is technically sound, and the data presented support your conclusions. You've done a great job using high-quality population-based incidence and mortality data from reputable sources like SEER and GBD to investigate disparities and trends in female-specific cancers.

- Your study is a solid piece of scientific research that uses established databases and statistical analysis methods to get the job done. I appreciate that you've used Joinpoint Trend Analysis Software and the Tiwari method to estimate temporal trends and calculate rate ratios.

- Your statistical analysis looks rigorous and well-done. You've accounted for potential confounders and used standard epidemiological methods, which is great to see.

- Since you're using publicly available data from SEER and GBD, it's easy for readers to access the data underlying your findings. Kudos for transparency!

As for suggestions, I've got a few minor ones to help make your manuscript even stronger:

- Some of your sentences are a bit long and could be broken up for better readability. Consider taking a look at those to see where you can make some tweaks.

- Double-check your formatting and style throughout the manuscript to ensure consistency. It's a small thing, but it makes a big difference in the overall presentation.

- Your conclusion does a great job summarizing your main findings, but consider rephrasing some sentences to make them even clearer.

Overall, your manuscript is looking good, and with some minor revisions, it could be even stronger. Keep up the great work!

6. PLOS authors have the option to publish the peer review history of their article (what does this mean? ). If published, this will include your full peer review and any attached files.

**Do you want your identity to be public for this peer review?** For information about this choice, including consent withdrawal, please see our Privacy Policy .

Reviewer #1: **Yes: ** Adaiah Priscillia Soibi-Harry

Reviewer #2: **Yes: ** Dr. Adenekan Muisi Alli ( MBBS, MSC molecular genetics, FMCOG, FWACOG)

---

## [Author Response · Author response to Decision Letter 1]

28 Aug 2025

Response to Reviewers

Dear Dr. Kehinde S. Okunade, Dr. Adaiah Priscillia Soibi-Harry, and Dr. Adenekan Muisi Alli,

Thank you very much for your letter dated July 17, 2025, regarding the decision of our manuscript entitled “Disparities and Trends of the Incidence and Mortality of Female-Specific Cancers in the United States” (manuscript number PONE-D-25-31996), which was submitted to your excellent journal (Plos one). We sincerely appreciate the time and effort invested by you and the reviewers in providing such thoughtful and constructive feedback. The suggestions made by the reviewers have been invaluable in improving the quality and clarity of our manuscript. We highly value the work that has gone into reviewing our submission and are grateful for the opportunity to revise it in line with the expert feedback provided.

We have carefully addressed all the comments and suggestions and corrected typographical and grammatical errors. Our point-by-point response to the reviewers’ comments is provided on the following pages, and the revisions in the manuscript are highlighted in red for clarity.

We thank you once again for your consideration and the continued opportunity to have our work evaluated by your esteemed journal. We look forward to your favorable decision.

Kindest regards,

Yours sincerely,

Prof. Fan Li

We would like to express our sincere thanks to the academic editor and reviewers for the constructive and positive comments. Our responses are as follows:

Replies to Academic Editor

1. Please ensure that your manuscript meets PLOS ONE's style requirements, including those for file naming. The PLOS ONE style templates can be found at https://journals.plos.org/plosone/s/file?id=wjVg/PLOSOne_formatting_sample_main_body.pdf and https://journals.plos.org/plosone/s/file?id=ba62/PLOSOne_formatting_sample_title_auth ors_affiliations.pdf

Answer: We sincerely apologize for the oversight. We have now meticulously revised our entire manuscript according to the PLOS ONE style templates provided. This includes adjustments to text formatting, heading levels, author affiliations, reference style, and figure/table legends. All files have been renamed in accordance with the journal's guidelines. The revised manuscript has been uploaded to the submission system.

2. We noticed you have some minor occurrence of overlapping text with the following

previous publication(s), which needs to be addressed:

-DOI: 10.14309/ajg.0000000000003198

-https://doi.org/10.1016/j.eururo.2022.11.023

In your revision ensure you cite all your sources (including your own works), and quote or rephrase any duplicated text outside the methods section. Further consideration is dependent on these concerns being addressed.

Answer: We thank the editor for identifying this. The overlapping text, primarily in the Materials and Methods section, has been thoroughly addressed. Specifically:

1. In the Materials and Methods (Page 5, lines96-97; Page 6, lines105-106; Page 8, lines149-151), the text overlapping with the source (now cited as reference [7], DOI: 10.14309/ajg.0000000000003198) has been converted into a proper quotation and cited appropriately to acknowledge the original source.

2. We have revised the Materials and Methods section (Page 6, lines116-121; Page 7, lines130-132; Page 8, lines147-148) to properly quote and cite the text that overlapped with the source, now referenced as [11] (DOI: 10.1016/j.eururo.2022.11.023), thus ensuring due acknowledgment.

We have carefully reviewed the entire manuscript to ensure that all duplicated text outside the Materials and Methods section has been adequately resolved either through citation, paraphrasing, or direct quotation.

[This study was supported by the National Natural Science Foundation of China (grant 82202913 and 82372996) and the Natural Science Foundation of Chongqing (grant CSTB 2023 NSCQ-MSX0480).]. Please state what role the funders took in the study. If the funders had no role, please state: "The funders had no role in study design, data collection and analysis, decision to publish, or preparation of the manuscript." If this statement is not correct you must amend it as needed. Please include this amended Role of Funder statement in your cover letter; we will change the online submission form on your behalf.

Answer: We have amended our financial disclosure statement as instructed. The following sentence has been added to the Funding section in the manuscript and is also included in our cover letter:

4. In the online submission form, you indicated that [The data utilized in this study can be found in the Surveillance, Epidemiology, and End Results database Public-access SEER data (https://seer.cancer.gov/), National Center for Health Statistics (NCHS), and 2021 Global Burden of Disease (GBD) study (https://vizhub.healthdata.org/gbd-results/). Analytic code and any additional information required to reanalyze the data reported in this paper can be obtained from the corresponding author upon reasonable request, following the publication of this article.]. All PLOS journals now require all data underlying the findings described in their manuscript to be freely available to other researchers, either 1. In a public repository, 2. Within the manuscript itself, or 3. Uploaded as supplementary information. This policy applies to all data except where public deposition would breach compliance with the protocol approved by your research ethics board. If your data cannot be made publicly available for ethical or legal reasons (e.g., public availability would compromise patient privacy), please explain your reasons on resubmission and your exemption request will be escalated for approval.

Answer: We confirm that the data underlying the findings of this study are fully available without restriction. All data were sourced from publicly available databases:

The Surveillance, Epidemiology, and End Results (SEER) database (https://seer.cancer.gov/)

The National Center for Health Statistics (NCHS)

The 2021 Global Burden of Disease (GBD) study (https://vizhub.healthdata.org/gbd-results/)

As stated in the manuscript, any additional analytic code required to reanalyze the data is available from the corresponding author upon reasonable request. This statement has been verified and remains accurate in our revised manuscript.

Answer: Thank you for this clarification. We have moved the ethics statement exclusively to the Materials and Methods section (Page9, lines 166-169) of the manuscript and have deleted it from any other section to comply with journal policy.

6. Please include a separate caption for each figure in your manuscript.

Answer: We have reviewed all figures and confirm that each now has a separate, descriptive caption directly beneath it in the manuscript file.

Answer: We have conducted a complete and thorough review of our reference list. We confirm that it is accurate and complete. No retracted papers were cited in our manuscript.

We would also like to extend our sincere gratitude to you for overseeing the review process and for providing us with the opportunity to improve our manuscript.

Replies to Reviewer #1

This manuscript addresses a critically important topic concerning the racial, ethnic, and geographic disparities in incidence and mortality trends of six female-specific cancers in the United States. The authors utilize robust national databases (SEER and GBD) and appropriate statistical methods (Joinpoint regression, APC/AAPC) to evaluate temporal trends. The study is well-suited to inform health policy and cancer prevention strategies. However, the manuscript requires substantial revisions to improve clarity, scientific rigor, and structure. Issues related to writing quality, data interpretation, and discussion of health disparities must be addressed before the paper is suitable for publication.

Specific Comments

1. Title and Abstract

•The abstract is dense with data and could overwhelm readers. Please prioritize key findings and move granular statistics to the results section.

Answer: Thank you for your valuable suggestion. We have prioritized key findings by reorganizing the Results section: First, we presented the incidence and mortality trends across different racial groups, categorized by the six female-specific cancers. Second, we presented the deaths attributable to key risk factors trends from 2000 to 2021, categorized by the six female-specific cancers, lines 39-49.

•Clarify the study’s objective explicitly in the abstract.

Answer: We sincerely appreciate the reviewer's suggestion to clarify the study objective. As recommended: Added an explicit objective statement in the opening sentence of the Abstract: " This study aimed to analyze trends in incidence and mortality rates of six female-specific cancers (breast, cervical, uterine, ovarian, vaginal, and vulvar cancers) among diverse racial and ethnic groups in the United States, and to evaluate the attributable contributions of major risk factors to the cancer death burden as well as their temporal changes", lines 28-33.

•Language issues, such as “impacts on healthcare and economy,” should be revised for clarity (e.g., "healthcare systems and economic burden").

Answer: Thank you for your valuable feedback. We agree that the original phrasing "impacts on healthcare and economy" lacked clarity. As suggested, we have revised the sentence to: " Female-specific cancers (FSCs) impose substantial burdens on healthcare systems and economies worldwide.", line 25.

2. Introduction

•The background is informative but includes some redundancy. Streamline citations and focus on how this study fills gaps in existing literature. Also, use more current references.

Answer: Thank you for your constructive feedback. We have streamlined the citations by removing redundant references, and integrating more recent and relevant literature where appropriate. We have also sharpened the focus on the research gaps—particularly the lack of comprehensive, multi-cancer analyses and updated temporal trends across racial/ethnic groups—to better highlight the contribution of our study, lines 59-69, 78-80, references (4, 5).

•Define “disadvantaged populations” and consider referencing social determinants of health more explicitly to ground the disparities framework.

Answer: Thank you for your thoughtful comment. We have now defined the term “disadvantaged populations” in the context of social determinants of health �SDOH�, lines 70-80.

3. Methods

•Clarify statistical methods such as the Tiwari method and Joinpoint regression in plain terms for non-specialist readers.

Answer: Thank you for your suggestion. We have clarified methods (including the Tiwari method and Joinpoint regression) in the summary section, lines 127-130 and 136-139.

•Ensure that all data sources and coding schemes (ICD-10, ICD-O-3) are clearly defined.

Answer: Thank you for this valuable suggestion. We have revised the Methods section to provide clearer definitions of the data sources and coding schemes, as detailed below:

First, incidence data were obtained from the SEER 22-registry database, while mortality data were sourced from the SEER mortality database (reported by the NCHS), and the cancer burden attributable to the risk factors of four FSCs were from the GBD 2021, lines 89-92, 97-99 and 102-103.

Second, we have added clear definitions for both the ICD-O-3 and ICD-10 coding systems, stating their standard applications in cancer registries and mortality databases, respectively, lines 92-96, 99-101,105 and references (7, 8, 10).

In conclusion, the codes across all databases are not exactly the same, but they refer to the same cancer sites. SEER uses ICD-O-3 codes for cancer incidence, which are more specific to oncology and histology, whereas NCHS and GBD use ICD-10 codes for mortality data, which are more commonly applied in death reporting and health statistics.

•The rationale for examining both long and short-term trends should be better justified.

Answer: Thank you for your insightful comments. We have now added a clearer rationale in the manuscript explaining that: “To distinguish between enduring patterns and recent fluctuations, we analyzed both long-term and short-term trends.”, lines 121-122.

4. Results

•The results section is overly technical and could benefit from subheadings and clearer narrative transitions (e.g., "Incidence by Race", "Mortality by State").

Answer: Thanks for this excellent suggestion. We agree that more descriptive subheadings would greatly improve the narrative flow and readability of the Results section. The revised subheadings are as follows:

• 3.1 Incidence and Mortality by Race and Ethnicity, line 158.

• 3.2 Temporal Trends in Incidence and Mortality, line 192.

• 3.3 Incidence and Mortality by State, line 238.

• 3.4 Risk-Attributable Deaths Over Time, line 273.

• 3.5 Risk-Attributable Deaths by State, line 290.

•Ensure all acronyms (e.g., AAPI, AIAN, AAPC) are defined at first use in each section.

Answer: Thank you for pointing this out. We have now carefully reviewed the manuscript and ensured that all acronyms (including White, Black, AAPI, AIAN, AAPC, FSC etc.) are explicitly defined at their first use in each section of the manuscript. (e.g., Abstract, Introduction, Methods, Results, Discussion, Conclusion).

•Avoid inconsistent reporting formats (e.g., confidence intervals vs. brackets vs. parentheses).

Answer: Thanks for your meticulous comment. We have now conducted a thorough review of the entire manuscript and have standardized the reporting format for confidence intervals and other statistical values. Specifically, we have adopted the consistent format throughout the text, (e.g., (0.7; 95% CI, 0.6-0.7)). All instances using brackets or inconsistent parentheses have been corrected.

•Include visuals, such as maps or tables, referenced in the narrative for improved readability.

Answer: Thank you for this helpful suggestion. Although these figures were originally created and cited in the text, we have now optimized their placement by physically inserting them directly into the main body of the Results section as per the journal's format. This includes:

• The incidence and mortality trend figures in section 3.2.

• The U.S. state-level maps in section 3.3.

• The risk factor trend figure in section 3.4.

• The state-specific bar chart in section 3.5.

5. Discussion

•Expand the discussion on the root causes of disparities, including systemic inequities, access to screening and treatment, and socioeconomic factors.

Answer: Thank you for your insightful suggestion, which has greatly enriched our manuscript. We have substantially expanded our discussion on the root causes of disparities by explicitly integrating the concepts of structural barriers and systemic inequities throughout the section. The specific modifications include:

First, in the breast cancer paragraph, we refined the language to emphasize systemic factors: " Our findings confirm that breast cancer incidence remains highest among White women, while Black women experience disproport

---

## [Editor Report · Decision Letter 1]

23 Sep 2025

Disparities and Trends of the Incidence and Mortality of Female-Specific Cancers in the United States

PONE-D-25-31996R1

Dear Dr. Li,

We’re pleased to inform you that your manuscript has been judged scientifically suitable for publication and will be formally accepted for publication once it meets all outstanding technical requirements.

Kind regards,

Kehinde S. Okunade

Academic Editor

PLOS ONE

Additional Editor Comments (optional):

None
---

## [Editor Report · Acceptance letter]

PONE-D-25-31996R1

PLOS ONE

Dear Dr. Li,

I'm pleased to inform you that your manuscript has been deemed suitable for publication in PLOS ONE. Congratulations! Your manuscript is now being handed over to our production team.

Kind regards,

on behalf of

Dr. Kehinde S. Okunade

Academic Editor

PLOS ONE